# Chemometric Profiling and Bioactivity of Verbena (*Aloysia* *citrodora*) Methanolic Extract from Four Localities in Tunisia

**DOI:** 10.3390/foods10122912

**Published:** 2021-11-24

**Authors:** Sonia Tammar, Nidhal Salem, Wissem Aidi Wannes, Hajer Limam, Soumaya Bourgou, Nedia Fares, Sarra Dakhlaoui, Majdi Hammami, Saber Khammassi, Giovanni Del Re, Kamel Hessini, Kamel Msaada

**Affiliations:** 1Laboratory of Aromatic and Medicinal Plants, Biotechnology Center in Borj-Cedria Technopol, BP. 901, Hammam-Lif 2050, Tunisia; tammar.sonia@hotmail.com (S.T.); hajerlimam39@gmail.com (H.L.); iyedazer2015@gmail.com (S.B.); saradakhlaoui@gmail.com (S.D.); hammamimajdi@gmail.com (M.H.); sabeurkhammassi@gmail.com (S.K.); msaada.kamel@gmail.com (K.M.); 2Faculté des Sciences de Bizerte, Université de Carthage, Zarzouna, Bizerte 7021, Tunisia; 3Laboratory of Bioactive Substances, Biotechnology Center in Borj-Cedria Technopol, BP. 901, Hammam-Lif 2050, Tunisia; salelnidhal@gmail.com (N.S.); hammamimajdi@hotmail.com (N.F.); 4College of Sciences of Tunis, Tunis El Manar University, Tunis 2092, Tunisia; 5Dipartimento di Ingegneria Industriale e dell’ Informazione e di Economia, Università dell’Aquila, Piazzale Ernesto Pontieri, Monteluco di Roio, 67100 L’Aquila, Italy; Giovanni.delre@gmail.com; 6Department of Biology, College of Sciences, Taif University, P.O. Box 11099, Taif 21944, Saudi Arabia; kamelhessini@gmail.com

**Keywords:** *Aloysia* *citrodora*, phenolic compounds, antioxidant activity, antibacterial activity, antifungal activity, anti-inflammatory activity, verbascoside, environmental factors

## Abstract

This research aimed to study the chemical composition of *Aloysia* *citrodora* methanolic extract and its biological activities as an antioxidant, and its antibacterial, antifungal and anti-inflammatory activities based on four bioclimatic collection stages. The contents of total phenols, total flavonoids and total tannins were determined. Nine phenolic compounds were identified by LC-DAD–ESI-MS/MS. The major compound was acteoside, a phenylpropanoid which represented about 80% of the methanolic fraction in the various regions. The antioxidant activities of different locations were measured by different analytical assays, such as DPPH, ABTS and iron reducing power. The results showed that phenolic compounds and antioxidant activities varied with climatic and environmental factors. Moreover, there was a significant dependency between regions and biological activities. The use of a principal component analysis showed that there was a close relationship among phenylpropanoids, phenolic compounds and the studied biological activities.

## 1. Introduction

Medicinal plants are an important source of secondary metabolites having therapeutic, cosmetic and food applications. Among the various plant protection products, phenolic compounds and essential oils generate a large group of secondary metabolites having a remarkable therapeutic effect and play the role of natural antioxidants and preservatives. In general, the effectiveness of the plant extracts depends on their yield, chemical composition and bioactivity. The latter could be generally affected by genetic and/or environmental factors [1]. Environmental factors, such as soil type, temperature, altitude and rainfall can affect the concentrations of phenolic compounds [2]. Therefore, phenolic compounds and other secondary metabolites represent a chemical interface between plants and the environment [3]. Variations in phenol quantity directly influence the quality of the plant for food products [4]. Food products rich in antioxidants are in demand because they act as scavengers of reactive oxygen species (ROS) [5].

Lemon verbena (*Aloysia citrodora*) belongs to the *Verbenaceae* family originating in South America and grows in North Africa, Southern Europe and various parts of Iran. Lemon verbena is one of the well-known aromatic species rich in phenylpropanoids and phenolic compounds [6] exerting the strongest antioxidant power [7]. Lemon verbena extract is extensively used in aromatherapy and perfumery applications. It is applied in traditional medicine for the cure of digestive problems, such as flatulence, indigestion and acidity [8]. The water infusion of verbena has been also ascribed for its antibacterial [9], antifungal [10], anti-inflammatory [11] and antioxidant [9] proprieties. Lemon verbena leaf extract is known for its high content of acteoside, named also verbascoside [12]. It has demonstrated potent biological effects, such as antioxidant, anxiolytic, neuroprotective, anticancer, anesthetic, antimicrobial and sedative properties [13]. Several studies reported that lemon verbena leaf extract contains a higher concentration of total phenols, tannins and flavonoids [14,15,16]. Wernert et al. [16] also described the presence of polyphenols (hydroxycinnamic acids and flavons) in lemon verbena leaf extract. On the other hand, various studies have been carried out on the antioxidant activity of the constituents of aqueous and alcoholic extract preparations of lemon verbena and antioxidant activity [17,18,19]. The aqueous and alcoholic extracts of the fresh aerial parts of Egyptian lemon verbena exhibited variable anti-inflammatory, antipyretic, analgesic and antioxidant properties [17]. However, there were few literature data concerning the effect of environmental conditions on the phenol composition and bioactivity of lemon verbena [20,21]. Rezig et al. [20] studied the variability of phenolics and the biological activities of Tunisian lemon verbena as affected by the geographic and climatic conditions of the collection areas. Chrysargyris et al. [21] investigated the fluctuation of polyphenols and antioxidant activities of Greece lemon verbena cultivated under two environmental conditions characterized by different altitudes.

In this way, this work was a comparative study of the variation of the chemical composition and antioxidant, antibacterial and antifungal potential among methanolic lemon verbena extracts extracted separately from four Tunisian climatic stages (Upper Arid, Middle Arid, Interior Sub-humid and Upper Semi-Arid). The detected variability was statistically interpreted and discussed according to environmental conditions of the studied sites underlining the environment’s effect on the diversify yield, chemical composition and biological activities of lemon verbena extract.

## 2. Materials and Methods

### 2.1. Plant Material

Lemon verbena aerial parts were randomly harvested in early to mid-October 2019 by hand from plants at the flowering stage cultivated in the Tunisian areas of Boussalem, Belli, Kairouan and Gabes (Table 1). The voucher specimen was deposited at the herbarium of our laboratory under the number “TC2011-2405”. The aerial parts of the lemon verbena were air-dried at room temperature for approximately seven days until reaching a constant mass of the plant material. Then, the lemon verbena aerial parts were ground and conserved at ~25 °C in darkness for further extractions.

### 2.2. Methanolic Extract Preparation

The air-dried samples were finely ground (grain size 0.7 mm) with a blade mill (IKA-WERK Type: A: 10). Triplicate 1 g subsamples of crushed aerial parts were extracted by shaking with 10 mL of pure methanol for 30 min at room temperature (25 °C). The extract was then stored for 24 h at 4 °C, filtered through Whatman No. 4 filter paper, evaporated in a vacuum (Rotavapor^®^ R-100, BUCHI, Flawil, Switzerland) to dryness and stored at 4 °C until analysis [22].

The extraction yield was 110 mg/g DW for Kairouan, 80 mg/g DW for Gabes, 60 mg/g DW for Boussalem and 180 mg/g DW for Belli. The extracts obtained will be used for the quantification of phenolic compounds and the evaluation of antioxidant activities.

### 2.3. HPLC-PDA-ESI-MS/MS Analysis

HPLC systems contained a photodiode array detector (PDA), a triple quadrupole mass spectrometer type Micromass Autospec Ultima Pt (Kelso, UK), and an ESI ion source working in negative mode was used for the identification of phenolic compounds. The latter was also done by a comparison of their retention time and fragmentation pattern with those of authentic standards and/or literature data [23,24].

### 2.4. Determination of Total Polyphenol, Flavonoid and Condensed Tannin Contents

The total polyphenol content was colorimetrically determined by using the Folin–Ciocalteu reagent [25,26]. The total polyphenol contents were expressed as mg of gallic acid equivalents per gram of dry weight (mg GAE/g DW).

The total flavonoid content was measured according to the method described by Kim et al. [27]. The absorbance of the mixture versus a prepared blank was read at 510 nm. These contents were expressed as mg of catechin equivalents per gram of dry weight (mg CE/g DW).

The total tannin content was measured using the modified vanillin assay described by Sun et al. [28]. The amount of total condensed tannins was expressed as mg (+)-catechin equivalent per gram of dry weight (mg CE/g DW).

Triplicates measurements were taken for total polyphenols, flavonoids and condensed tannins.

### 2.5. Antioxidant Activity

The electron donation ability of lemon verbena methanolic extract in relation to the growing region was determined using the DPPH assay according to Hatano et al. [29]. The antiradical activity was expressed as IC_50_ (μg/mL), the concentration required to cause 50% DPPH inhibition. BHT was used as the positive control.

The ABTS radical cation (ABTS^+^) was performed by the method described by Re et al. [30]. The ABTS scavenging capacity was expressed as IC_50_ (µg/mL), and BHT was used as the positive control.

The reducing power of methanolic lemon verbena extract was determined according to Amezouar et al. [31]. Ascorbic acid was used as the positive control. The results are expressed as IC_50_ values (µg/mL).

### 2.6. Antimicrobial Assay

Six bacterial strains: Staphylococcus aureus ATCC 6538, Listeria monocytogenes ATCC 19115, Enterococcus faecalis ATCC 29212, Escherichia coli ATCC 25922, Pseudomonas aeruginosa ATCC 2134 and Salmonella arizonae DMB 560 and three yeast strains: Candida albicans ATCC 10231, Fusarium solani and Rhizoctonia solani were taken from the microorganism collection of our laboratory.

The disc diffusion method was used to determine the antibacterial activity by measuring the diameter of the growth inhibition zone (IZ) around the discs as described before by Rios and Recio [32]. The mycelial growth inhibitory effect was performed as described by Rguez et al. [33]. The minimum inhibitory concentration was determined by the microdilution method [34]. Gentamicin and nystatin were used as positive standards.

### 2.7. Anti-Inflammatory Activity

The cell line (RAW 264.7 murine macrophage), supplied from the American Type Culture Collection (ATCC, Manassas, VA, USA) was cultured and kept at 37 °C in a humidified environment of 5% CO_2_ [35].

To evaluate the in vitro cytotoxicity of the methanolic lemon verbena extract [36], RAW macrophage cells were seeded in 24-well plates at 5 × 10^4^ cells/well and allowed to attach. Then, the cells were treated with increasing concentrations for 24 h. The proportionality of stained cells with viable cell numbers at the absorbance at 540 nm was determined. Cell viability in each well was calculated as a % of the control. The Resazurin reduction test was used to determine the extract cytotoxicity [37].

To determine the quantity of the nitrite production, RAW 264.7 cells were seeded in 24-well plates at a density of 2 × 10^5^ for 24 h at 37 °C. Then, after 60 min, cells were treated with lemon verbena extracts, dissolved in the DMSO and stimulated with 100 mg/mL lipopolysaccharide (LPS). After 24 h, the quantity of nitrite accumulated in the culture supernatant was determined [38].

### 2.8. Statistical Analysis

All results of these assays were repeated three-fold and expressed as mean values and standard deviation. A one-way and multivariate analysis of variance (ANOVA) followed by Duncan’s multiple range test and Tukey’s test were performed by SPSS 15 (SPSS Inc. Chicago, IL, USA), Pearson and PCA tests were performed by XLSTAT (XLSTAT Version 2014.5.03 Copyright Addinsoft, Paris, France).

## 3. Results and Discussion

### 3.1. Methanolic Extract Composition

#### 3.1.1. Identification of Phenolic Compounds

The fingerprinting of methanolic lemon verbena extract led to the detection of nine phenolic compounds (Table 2).

Peak 1 (Rt 11.17 min; λ_max_, 326 nm) exhibited a pseudomolecular ion [M-H]^−^ at *m*/*z* 335, yielding an intense fragment ion at *m*/*z* 179 (caffeoyl residue) through the loss of shikimic moiety detected as caffeoylshikimic acid [24].

Peak 2, eluted at 14.08 min, presented a pseudomolecular ion [M-H]^−^ at *m*/*z* 305 and a fragment ion at *m*/*z* 289 indicative of the catechin or (*epi*)-catechin aglycone. This compound was tentatively identified as a catechin- or an (*epi*)-catechin gallate [39].

Peaks 3 and 4 (λ_max_, 326 nm), eluted at *t_R_*_,_ 16.02 and 16.7 min, respectively displayed a pseudomolecular ion [M-H]^−^ at *m*/*z* 515 with an intense MS^2^ fragment ion at *m*/*z* 353 (caffeoylquinate residues). This fragmentation pattern was consistent with Clifford et al. [23] for 3,4-di-caffeoylquinic and 3,5-di-caffeoylquinic acids.

Peaks 5, 6 and 7 showed a characteristic UV spectrum of phenylethanoid glycosides with absorption peaks at 249, 289 and 331 nm. Peak 5 (Rt 24.26 min) exhibited a pseudomolecular ion [M-H]^−^ at *m*/*z* 623, and its fragmentation gave an intense fragment ion at *m*/*z* 461 [M-H-162]^−^ (loss of caffeoyl moiety). This component was the most abundant one, and it was identified as acteoside, also known as verbascoside [40]. Peak 6 showed similar UV (λ_max_, 249, 289 and 328 nm) and mass spectral ([M-H]^−^ at *m*/*z* 623; MS^2^ at *m*/*z* 461) characteristics, with peak 5 suggesting that both compounds (peaks 5 and 6) were isomers. Consequently, compound 6 was tentatively identified as an isoacteoside (isoverbascoside). Peak 7 (Rt 27.83 min; λ_max_, 281, 325 nm) presented a pseudomolecular ion [M-H]^−^ at *m*/*z* 651, and an intense fragment at *m*/*z* 475 was owed to the removal of feruloyl moiety (176 amu) and identified as martynoside [41].

Peak 8 (Rt 39.44 min) showed a characteristic UV spectrum of flavone (λ_max_, 337 nm), exhibited a deprotonated molecular ion [M-H]^−^ at *m*/*z* 299 and was tentatively identified as diosmetin.

Peak 9 (Rt 40.72 min; λ_max_, 339 nm) showed similar mass spectral data ([M-H]^−^ at *m*/*z* 299) to that observed for the commercial standard apigenin.

#### 3.1.2. Phenolic Composition of Methanolic Lemon Verbena Extracts

According to Table 3, the phenolic composition of the methanolic lemon verbena extracts was influenced by the bioclimatic stage effect. These extracts were characterized by a high acteoside content. The methanolic extract from Boussalem region was the richest in acteoside with a rate of 83.54%, followed by that extracted from the Kairouan region with a percentage of 81.94%, the other two regions were marked by 76.07% for Belli and 76.15% for Gabes. This variation was statistically significant with a *p*-value equal to 0.000.

Acteoside (also known as verbascoside) is a phenylpropanoid widely distributed in the plant kingdom and especially in the *Aloysia* species [12,41]. The second major compounds were 3,4-Di-caffeoylquinic acid, isoacteoside, martynoside and diosmetin with significantly varied percentages. The Belli region was marked by the highest levels of 3,4-Di-caffeoylquinic acid (4.81%) and diosmetin (4.58%). 3,4-Dicaffeoylquinic acid (3,4-Di-O-caféoylquinic acid), naturally isolated from *Laggera alata*, has antioxidant, DNA protective, neuroprotective and hepatoprotective properties. 3,4-Dicaffeoylquinic acid exerts apoptosis-induced cytotoxicity and glucosidase inhibitory effects [42].

Diosmetin is a bioflavonoid abundant in citrus fruits, legume leaves and spermine. It exerts antioxidant, anti-inflammatory, antiapoptotic, antimutagenic and antibacterial effects. Likewise, this flavone promotes strong cellular antioxidant activity in human monocytes by preventing the generation of intracellular ROS and the formation of malondialdehyde and by increasing the effects of the intracellular antioxidant enzymes superoxide dismutase, catalase and glutathione peroxidase. [43]. Isoacteoside is a phenylethanoide glycoside isomer of acteoside, its percentage was 4.19% for Gabes and 4.70% for Boussalem having the same biological activities (Figure 1).

The phenolic compounds analyzed by LC-DAD were subjected to a hierarchical ascending classification (HAC) of the lemon verbena populations according to the composition of phenolic compounds and climatic conditions which grouped regions according to their similar composition. As shown in Figure 2, the dendrogram sorted the regions into two main groups, namely A1 and A2. The first group A1 consisted of two lemon verbena populations located in the Upper Semi-Arid and Interior Sub-Humid regions. The second (A2) grouped together four populations located in the Upper Arid and Semi-Arid regions. This distribution was visibly influenced by the rainfall and soil (Table 1) which could have caused oxidative stress in plant species by producing antioxidants. Phenolic compounds are among the main substances responsible for antioxidant activities.

### 3.2. Phenolic Contents of Methanolic Lemon Verbena Extracts

From Table 3, lemon verbena had an appreciable content of total phenols ranging from 29.16 mg GAE/g for Boussalem to 11.66 mg GAE/g for Gabes. Similar results were obtained by Rezig et al. [20] who reported that the content of the total polyphenols of Tunisian lemon verbena varied from 14.25 to 26.58 mg GAE/g. Lemon verbena extracted with a phosphate buffer had a lower total phenol content of 1.55 mg GAE/g [44]. Likewise, Cheurfa and Allem [45] and Choupani et al. [46] determined the effect of the extraction solvent on the content of the total polyphenols of lemon verbena extracts. The highest total phenol content was observed in the methanolic extract having 25.94 mg GAE/g [46].

In descending order, the total flavonoid content was 39.86 mg CE/g in Boussalem, 38.86 mg CE/g in Kairouan, 37, 20 mg CE/g in Gabes and 27.53 mg CE/g in Belli. Lower results were detected by Rezig et al. [20] who reported that the content of total flavonoid of Tunisian lemon verbena varied from 10.59 to 19.26 mg CE/g using methanol as a solvent. Additionally, the aqueous extract of *Aloysia triphylla* had a weak content of total flavonoids (0.43 mg CE/g) as reported by Vinha et al. [47].

The lemon verbena of Kairouan and Boussalem showed the highest condensed total tannins (0.04 mg EC/g DM), followed by that of Belli (0.03 mg EC/g) and Gabes (0.02 mg EC/g). A higher content of total condensed tannins was observed by Rezig et al. [20] in the case of Tunisian lemon verbena which varied from 0.89 to 1.10 mg CE/g.

### 3.3. Antioxidant Activity

The results of the antioxidant capacities of the four collection regions are presented in Table 4. The methanolic extract of the Boussalem sample showed the highest DPPH activity (IC_50_ = 12.71 µg/mL) and the lowest in the Gabes sample (14.90 µg/mL). BHT had an IC_50_ = 17.00 µg/mL. Similar results were obtained by Rezig et al. [20] in the case of Tunisian lemon verbena. The aqueous extract of Algerian lemon verbena leaves had a lower antiradical activity (IC_50_ = 27.4 mg/mL) than that of the hydroalcoholic extract (IC_50_ = 23.52 mg/mL) [44]. A decoction of verbena (*Verbena officinalis* L.) had an IC_50_ = 15.76 mg/mL [48]. Likewise, the lemon verbena of the Boussalem region was characterized by a notable activity for ABTS (IC_50_ = 4.54 µg/mL) and a reducing power (IC_50_ = 10.37 µg/mL).

### 3.4. Antibacterial Activity

As can be seen in Table 5, the zone of inhibition diameter (IZ) was strongly influenced by the area factor for *S. aureus, L. monocytogenes, P. aeuroginosa* and *S. arizonae* (*p* < 0.001). In contrast, the methanolic lemon verbena extracts did not show a significant difference in antibacterial activity against *E. coli*. The highest antibacterial activity was observed against *L. monocytogenes* showing an inhibition zone equal to 32.5 mm in methanolic extracts of lemon verbena from Kairouan, followed by Gabes (IZ = 17.33 mm) against *S. arizonae*, Kairouan (IZ = 16.33 mm) against *E. faecalis* and Boussalem (IZ = 14 mm) against *P. aeuroginosa*. The lower antibacterial activity of methanolic lemon verbena extracts was measured against *E coli* for the Belli region.

Similar results were found by Kumar et al. [49] who studied the antibacterial activity of aqueous and organic extracts of lemon verbena aerial parts against Gram-positive (*B. subtilis*, *S. aureus*) and Gram-negative (*E. coli*, *K. pneumoniae* and *P. vulgaris*) bacteria. They found that all the bacterial strains were resistant to the aqueous extract, while the methanol and ethanol extracts showed moderate activities (IZ = 10.0–32 mm). A slight inhibition of growth was observed with extracts of chloroform, diethyl ether and chloroform-methanol (3:1) against all these studied strains (IZ = 6.2–8.4 mm). The phenolic compounds and phenylpropanoids were responsible for the antibacterial effect by the inactivation of microbial adhesins. The transport proteins and cell envelope may be related to the mechanism of inhibition of hydrolytic enzymes, such as proteases, carbohydrases and others interactions [41]. Indeed, the major compound of this extract was acteoside (verbascoside), this compound had significant antibacterial activity. Bazzaz et al. [50] reported that verbascoside and caffeine were able to decrease the MIC of gentamicin against standard resistant strains and two clinical isolates, *Staphylococcus aureus* and *Escherichia coli*. Among these associations, the co-administration of verbascoside and gentamicin was more effective, and synergistic activities (FICI < 1) against clinical isolates were observed.

### 3.5. Antifungal Activity

The antifungal activity evaluation consisted in measuring the percentage of the mycelial growth inhibition of *C. albicans*, *F. solani* and *R. solani* in the presence of methanolic lemon verbena extracts (Figure 3).

The methanolic lemon verbena extracts had a significant variation in antifungal activity. The results revealed that *F. solani* and *R. solani* were the strains most affected by the methanolic fractions extracted from the regions of Boussalem and Kairouan, where the percentages of inhibition reached 100% at the dose of 500 mg/L. *C. albicans* appeared less sensitive than the two previous strains, the extract from Kairouan was the most active on this strain with a dose of 2000 mg/L. The methanolic lemon verbena extract was characterized by a high content of phenylpropanoid glycoside, acteoside (verbascoside) and its isomers isoacteoside and martynoside (about 90%). These molecules are known for their important biological activities including antibacterial and antifungal potential. In fact, Ali et al. [51] demonstrated that acteoside significantly reduced the inhibitory concentration of amphotericin B against clinically important fungal pathogens (the *Cryptococcus neoformans*, *Candida* species and *Aspergillus* species). Nikonorova et al. [52] proved that verbascoside inhibited the in vitro growth of *F. culmorum*, *B. sorokiniana* and *B. cinerea* but not of *R. solani* and *A. solani*. At low concentrations, verbascoside influenced the morphological development of pathogenic fungi.

### 3.6. Anti-Inflammatory Activity

The relative positions of the lemon verbena collection regions according to their phenolic composition using a hierarchical ascending classification showed that the methanolic lemon verbena extracts were grouped into two clusters (A1: Belli/Boussalem and A2: Gabes/Kairouan) according to their similarities in composition in phenolic compounds (Figure 1). In fact, a region of each cluster, namely Kairouan and Boussalem, was selected to determine the anti-inflammatory activity of the methanolic extracts. In addition, according to the results of the activities and assays carried out previously, these two regions are characterized by an important antioxidant potential. The cytotoxicity effect of methanolic lemon verbena extracts was evaluated (Figure 4).

The RAW 267.4 cells were treated with increased extract concentrations (1.25 to 5 µg/mL). The extracts showed no significant cytotoxicity against the RAW 267.4 macrophage cells up to 5 µg/mL. Indeed, our experiments were carried out at nontoxic concentrations of 1.25, 2.5 and 5 µg/mL. In general, inflammation is a natural response that initiates the destruction of pathogens, tissue repair processes and restoration of homeostasis to infected or damaged sites [53]. The effect of the methanolic extracts from Kairouan and Boussalem on the production of NO • is illustrated in Figure 5.

The results show that the various lemon verbena extracts exhibited a significant anti-inflammatory activity. The extracts from the Boussalem region showed the best activity inhibiting the production of NO • at an IC_50_ = 2.74 µg/mL, the lemon verbena collected in Kairouan gave an activity of an IC_50_ = 3.39 µg/mL. In line with our results, Lenoir et al. [54] showed that the preventative consumption of lemon verbena infusion could help manage oxidative stress by stimulating antioxidant enzymes, such as SOD, and reducing lipid peroxidation in a colonic inflammation model in rats.

### 3.7. Principle Component Analysis

To better understand the climatic effects on the composition and the biological activities of the methanolic lemon verbena extracts, a principal component analysis allowed the analysis of the interactions between phenolic compounds and the results of biological activities (Figure 6).

The variability percentages associated with the axes of the representation space were presented by F1 (52.53%) and F2 (25.12%) with a sum of 77.65%, which shows that the representation is reliable. The region distribution showed that Gabes and Belli were statistically very close, with the highest IC_50_ of antioxidant activities (DPPH, ABTS and reducing power). However, they were characterized by bactericidal activity against *S. aureus, E. coli* and *S. arizonae*. These interpretations were approved by a similarity in phenolic composition for diosmetin and apigenin (4.58% for Belli and 2.35% for Gabes). Similar results were reported by Nayaka et al. [55], Morimoto et al. [56] and Cheng et al. [57] which proved the antibacterial potential of these flavones. The Upper Arid stage represented by Kairouan was marked by martynoside (8.18%) and catechin-galtate (1.01%). The lemon verbena extract from this region was characterized by antibacterial activity against *L. monocytogen* and *E. faecalis*, as well as antifungal activity against *R. solani* and *C. albicans*. According to Pendota et al. [58], martynoside is a phenylpropanoid glycoside isolated from the butanol fraction of *Boscia albitrunca*, it was the most active with the lowest MIC values of 7.81 and 31.2 µg/mL against *B. subtilis* and *K. pneumoniae,* respectively. However, it showed moderate activity against *C. albicans* with MIC values ranging from 62.5 to 250.0 µg/mL. The Lower Sub-Humid stage, represented by the Boussalem region, was associated with the highest mean rainfall factor. A chromatographic analysis of the methanolic extract from Boussalem showed that it was the richest in phenylpropanoid glycoside. These results were consolidated by Mechri et al. [59] and Mechri et al. [60]; Verbascoside and oleuropein were potential indicators of drought resistance in olive trees (*Olea europaea* L.). Although Boussalem and Kairouan have different bioclimatic stages, they are located at a close altitude (143 and 122 m, respectively). It is possible that this factor explained their similarity in phenylpropanoid glycoside contents and consequently a similarity in antioxidant activity. Chrysargyris et al. [20] found that the altitude factor affected the antioxidant capacity of lemon verbena, as plain plants presented higher flavonoids and DPPH than mountainous grown plants. This observation was consolidated by the correlation matrix (Pearson (*n*)) which studied the statistical interaction between composition and biological activities. This study clearly showed that acteoside had a significant correlation with the IC_50_ of antifree radical activities DPPH and ABTS and the iron reducing power with very remarkable coefficients: −0.988, −0.980 and −0.992. The antioxidant potential of acteoside and isoacteoside has been shown by several previous studies [61,62]. Likewise, Vertuani et al. [63] showed that the addition of verbacoside increases the oxidative stability of cosmetic formulas.

## 4. Conclusions

In conclusion, the analysis of the biochemical variability of Tunisian verbena species showed that the chemical composition of methanolic extracts is similar to variations in the amounts of each compound identified. This trend was also recorded for all biological activities; the best data are associated with the region of Boussalem, this region is located in a lower Sub-Humid stage with an important rainfall and an average altitude. Antioxidant activity is correlated with the amount of phenylpropanoid including acteoside and isoacteoside. The other activities have no significant correlation with climatic factors. Regardless, these extracts are endowed with a remarkable antibacterial, antifungal and anti-inflammatory power, which shows that this extract can play the role of an important cosmetic additive to increase shelf life dates.

## Figures and Tables

**Figure 1 foods-10-02912-f001:**
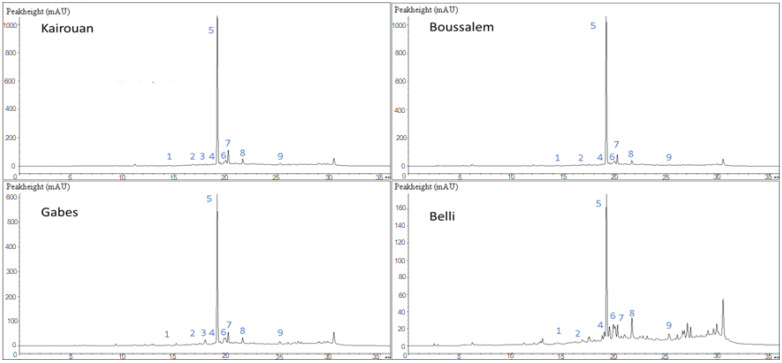
Chemical composition chromatographic profile of methanolic lemon verbena extracts depending on the location of collection by LC-DAD. The affiliation of the numbered peaks is reported in Table 3.

**Figure 2 foods-10-02912-f002:**
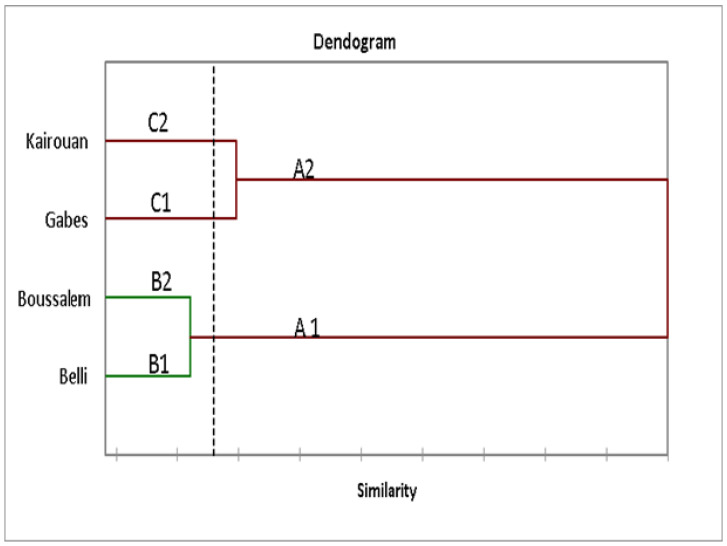
Relative positions of lemon verbena collection regions according to their phenolic composition using a hierarchical ascending classification (HAC).

**Figure 3 foods-10-02912-f003:**
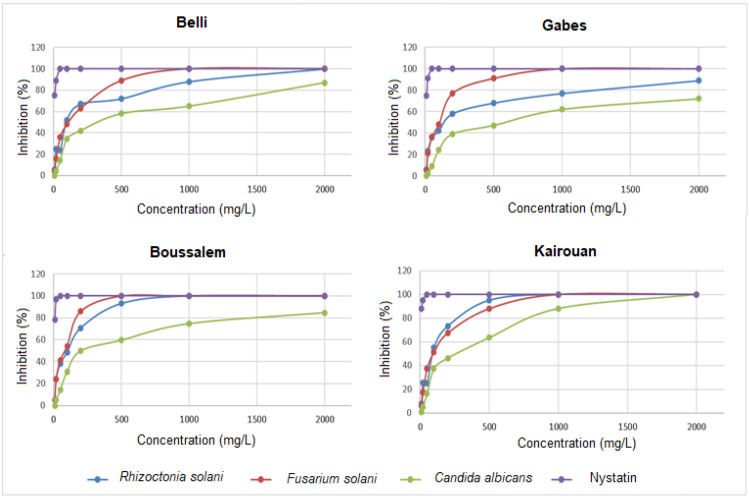
Variation in the percentages of inhibition of *Rhizoctonia solani, Fusarium solani, Candida albicans* and Nystatin depending on the location of collection. The values are the average of three replicates.

**Figure 4 foods-10-02912-f004:**
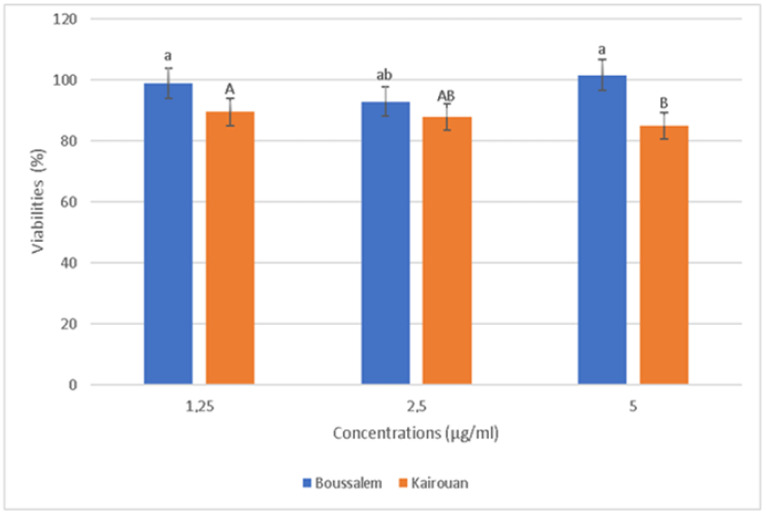
Evolution of cell viability as a function of the concentrations of lemon verbena extracts in different collection regions. Percentages with different letters (a,A,b,B) were significantly different at *p* < 0.05.

**Figure 5 foods-10-02912-f005:**
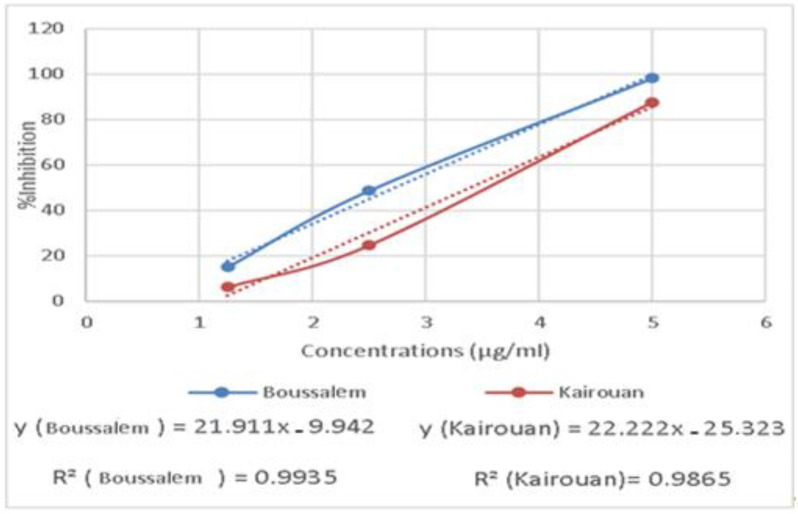
Variation of anti-inflammatory activity of the methanolic lemon verbena extracts depending on the location of collection.

**Figure 6 foods-10-02912-f006:**
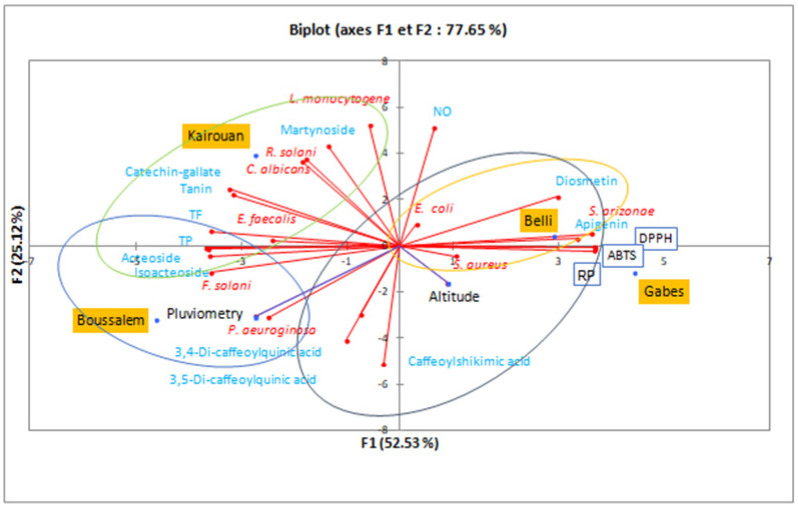
Principal component analysis in a biplot of the lemon verbena populations according to the composition of methanolic extracts, their activities and climatic conditions. TP: total polyphenols, TF: total flavonoids, NO: anti-inflammatory, DPPH, ABTS: antifree radical activity, RP: reducing power.

**Table 1 foods-10-02912-t001:** Studied area data of Tunisian lemon verbena.

	Bioclimatic Zone	Longitude	Latitude	Elevation (m)	Mean Rainfall (mm/year)	Soil
Kairouan	Upper Arid	10°05′46″ E	36°27′21″ N	122	287	Sandy
Gabes	Middle Arid	10°05′53″ E	33°52′53″ N	299	223	Sandy
Boussalem	Interior Sub-Humid	8°46′48″ E	36°30′04″ N	143	537	Clay
Belli	Upper Semi-Arid	10°44′15″ E	36°27′21″ N	14	326	Clay

**Table 2 foods-10-02912-t002:** Retention time (Rt), spectral data and tentative identification of methanolic lemon verbena extract.

Peak	Rt (min)	λmax (nm)	[M-H]^−^ (*m*/*z*)	Production (*m*/*z*)	Tentative Identification
1	11.17	326	335	179	Caffeoylshikimic acid
2	14.08	273	305	289	Catechin-gallate
3	16.02	326	515	353	3,4-Di-caffeoylquinic acid
4	16.7	326	515	353	3,5-di-caffeoylquinic acid
5	24.26	249, 289, 331	623	461	Acteoside
6	25.46	249, 289, 331	623	461	Isoacteoside
7	27.83	281, 325	651	475	Martynoside
8	39.44	337	299	-	Diosmetin
9	40.72	339	269	-	Apigenin

**Table 3 foods-10-02912-t003:** Variation in the chemical composition of methanolic lemon verbena extracts depending on the collection area.

		Peak Area (%)				
	Phenolic Compound	Belli	Gabes	Boussalem	Kairouan	*p*
1	Caffeoylshikimic acid	0.64 ^c^ ± 0.03	0.85 ^b^ ± 0.04	0.95 ^a^ ± 0.05	0.53 ^d^ ± 0.03	0.001 ***
2	Catechin-gallate	0.88 ^b^ ± 0.04	0.86 ^b^ ± 0.04	0.96 ^ab^ ± 0.05	1.01 ^a^ ± 0.05	0.034 *
3	3,4-di-caffeoylquinic acid	4.81 ^a^ ± 0.24	1.05 ^b^ ± 0.05	4.85 ^a^ ± 0.24	0.86 ^b^ ± 0.04	0.001 ***
4	3,5-di-caffeoylquinic acid	1.28 ^b^ ± 0.06	0.86 ^c^ ± 0.04	1.55 ^a^ ± 0.08	0.67 ^c^ ± 0.03	0.001 ***
5	Acteoside	76.07 ^b^ ± 3.8	76.15 ^b^ ± 3.81	83.54 ^a^ ± 4.18	81.94 ^a^ ± 4.1	0.001 ***
6	Isoacteoside	4.35 ^ab^ ± 0.22	4.19 ^b^ ± 0.21	4.70 ^a^ ± 0.24	4.55 ^a^ ± 0.23	0.021 *
7	Martynoside	5.32 ^bc^ ± 0.27	5.85 ^b^ ± 0.3	5.59 ^b^ ± 0.28	8.18 ^a^ ± 0.41	0.001 ***
8	Diosmetin	4.58 ^a^ ± 0.23	4.04 ^ab^ ± 0.2	3.21 ^c^ ± 0.16	3.83 ^b^ ± 0.19	0.01 **
9	Apigenin	1.66 ^b^ ± 0.08	2.35 ^a^ ± 0.12	1.05 ^bc^ ± 0.05	1.44 ^b^ ± 0.07	0.001 ***
	Total flavonoids (mg CE/g DW)	27.53 ^b^ ± 1.03	37.20 ^a^ ± 1.36	39.86 ^a^ ± 0.64	38.86 ^a^ ± 0.49	0.000 ***
	Total Polyphenols (mg GAE/g DW)	12.75 ^c^ ± 0.13	11.66 ^c^± 0.12	29.16 ^a^ ± 0.30	25.50 ^b^ ± 0.26	0.000 ***
	Tannins (mg CE/g DW)	0.03 ^ab^ ± 0.01	0.02 ^c^ ± 0.01	0.04 ^a^ ± 0.01	0.04 ^a^ ± 0.01	0.002 **

The values shown in this table are the mean of three replicates and are given as the mean ± SD (*n* = 3). One-way ANOVA followed by Duncan’s multiple range test were used. The values with different exponents (^a–d^) were significantly different at *p* < 0.05. * *p* < 0.05; ** *p* < 0.01; *** *p* < 0.001.

**Table 4 foods-10-02912-t004:** Antioxidant properties against DPPH and ABTS radicals and the reducing power (RP) of methanolic lemon verbena extract.

	Collecting Regions				Synthetic Standard		*p*
	Belli	Boussalem	Gabes	Kairouan	BHT	Ascorbic Acid	
DPPH IC_50_ (µg/mL)	14.52 ^b^ ± 1.01	12.71 ^d^ ± 0.24	14.90 ^a^ ± 0.83	13.13 ^c^ ± 0.91	17 ± 0.41	-	0.001 ***
ABTS IC_50_ (µg/mL)	7.62 ^ab^ ± 1.05	4.54 ^c^ ± 1.13	8.10 ^a^ ± 1.12	5.61 ^c^ ± 0.92	16 ± 1.18	-	0.001 ***
RP IC_50_ (µg/mL)	15.24 ^b^ ± 1.06	10.37 ^b^ ± 1.33	16.02 ^a^ ± 1.15	11.24 ^c^ ± 1.07	-	4 ± 0.12	0.001 ***

IC_50_ values represent the mean of three replicates (*n* = 3); the letters (^a–d^) indicate significant differences at *p* < 0.05. *** Significant at *p* < 0.001%.

**Table 5 foods-10-02912-t005:** In vitro antimicrobial activities of methanolic lemon verbena extracts collected in four localities.

		Regions				Antibiotic	
		Belli	Gabes	Boussalem	Kairouan	Gentamicine	*p*
Gram +							
*S. aureus*	IZ	13 ^a^ ± 2	10.33 ^bc^ ± 1	11 ^b^ ± 1	10.33 ^bc^ ± 2	22 ± 2	0.001 ***
	MIC	0.25 ± 0.20				0.05 ± 0.01	
*L. monocytogenes*	IZ	26 ^b^ ± 1	25 ^bc^ ± 2	23 ^c^ ± 2	32.5 ^a^ ± 2	39 ± 3	0.001 ***
	MIC				0.92 ± 0.06	0.05 ± 0.01	
*E. faecalis*	IZ	12 ^b^ ± 1	15 ^a^ ± 2	16 ^a^ ± 2	16.33 ^a^ ± 1	15 ± 1.85	0.04 *
	MIC				1 ± 0.23	0.05 ± 0.01	
Gram −							
*E. coli*	IZ	11 ^a^ ± 1	9.5 ^a^ ± 1	10 ^a^ ± 1	10.33 ^a^ ± 1	29 ± 2	0.52 ^NS^
	MIC	0.43 ± 0.01				0.1 ± 0.42	
*P. aeuroginosa*	IZ	9 ^c^ ± 1	11.33 ^b^ ± 1	14 ^a^ ± 1	11.5 ^b^ ± 1	19 ± 2.21	0.001 ***
	MIC			1.2 ± 0.38		0.86 ± 2.21	
*S. arizonae*	IZ	15.5 ^ab^ ± 2	17.33 ^a^ ± 1	11 ^c^ ± 1	13 ^b^ ± 1	27.01 ± 2.7	0.001 ***
	MIC		1.33 ± 0.15			0.85 ± 0.76	

MIC: minimal inhibitory concentration (µg/mL), IZ: inhibition zone (mm). Values are given as the mean ± SD (*n* = 3) and are the average of three replicates. One-way ANOVA followed by Duncan’s multiple range test were used. Values with different superscripts (^a–c^) are significantly different at *p* < 0.05. Values followed by a common letter in columns are not significant (*p* > 0.05). NS: not significant. * significant at *p* < 0.1, *** significant at *p* < 0.001.

## Data Availability

Not applicable.

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
