# Peer review of "Chemometric Profiling and Bioactivity of Verbena (Aloysia citrodora) Methanolic Extract from Four Localities in Tunisia"

_foods, 2021, doi:10.3390/foods10122912_

Round 1
Reviewer 1 Report
The manuscript under appreciation is about the study of the chemical composition and bioactivity of Verbena.
The manuscript is interesting and provides novelty.
The following comments are to be taken into account by the authors:
In the “Introduction” section, line 21: the word “phenolic” should be added in order to be more specific. “Nine phenolic compounds were identified by…”
In the “Materials and Methods” section, line 61: please be more specific about the number of samples used in this study.
Regarding the identification of phenolic compounds in line 156: Did the authors identify several compounds based on available phenolic standards? In this case, they must report which compounds were identified based on standards, and which were identified based solely on literature data. The vendor and the purity should also be reported for standards.
Line 146: in order to perform ANOVA the data must follow a normal distribution. Did the authors perform a normality check?
In Table 2 and Figure 3 the legends should be on the same page.
In Table 3 the units for the nine compounds (% area?) are not reported. The same applies to Table 4 where the IC50 units are not shown properly.
Lines 222-223: Please rephrase, it does not make sense.
Line 242: “…..higher tha reported…” correct “tha” to “than”
Author Response
Dear Editor
You find as an attachment the response to reviewer.
Best Regards

Reviewer 2 Report
The manuscript titled “Chemometric Profiling and Bioactivity of Verbena (Aloysia citrodora) Methanolic Extract From Four Localities in Tunisia” is related to evaluation bioactivity of verbena plant collected at 4 different locations. The manuscript has several major drawbacks:
- There is very low information and conclusions about the environmental influence on plant chemical composition which was one of the goals of the study
- Many articles addressing these topics (but not for Tunisia) were already published
- Materials and methods should be explained in more details
Detail and minor issues are listed below
Lines 98-98 – bioactivity and bioavailability
- Section - some of the environmental influences on verbena should be enlisted, and some of the mechanisms should be mentioned so it could justify the necessity for this type study. There are many articles addressing biological activities and phenolics determination for verbena. Author should make a mini-review of what was already performed in the field and emphases the novelty of their work.
2.1 Section - Which days were selected for harvest and why those specific dates? In which vegetative phase were the collected plants? Were plant materials from 3 different harvests mixed and presented as one sample? How long was the air-dying process and where it was performed? Which was the water content after drying process?
2.2. Section – Please provide extraction yield for all plants.
Line 72 – vacuum
Line 72 – please provide model, producer of vacuum evaporator
Line 74 – and other in vitro bioactive tests, please rephrase the sentence
Sections 2.4-2.9. – Even though a reference was added for all assays a brief protocol of all assays should be written because for example for ABTS it is unclear which was the incubation time. Details would help scientific community to reproduce assays (if necessary) and make results comparable.
Section 2.10 Duncans test was not mentioned, please do so.
Line 161 and 164 – Peak
Table 3 - Units for TF and T are incorrect, please correct them. Are numbers appointed to individual phenolic peak areas, please add this information in table 3 and in text. Why at least for Acteoside concentration was not determined by analytical standard? Symbols for *, **, *** are not explained in the table legend, please do so. It is not correct to say significance is 0.000 (e.g. for Acteoside) but rather ˂0.001, please correct it.
Line 220 – I suppose Interior instead of Inferior, if so correct in the entire manuscript
Lines 220-224 – Which was the annual rainfall for all harvest locations? Before giving a statement that water scarcity could influence phenolics content variability of annual rainfall should be checked. Hawary et al. [32] evaluated the essential oil composition of Lippia citriodora in June and October, but did not mention the water availability. I didn’t check Cheurfa and Allem [33] 221 and Oukerrou et al. [34] but this entire part should be rewritten and stronger evidence for plant water availability and its influence on plants and synthesis of polyphenols should be presented.
Lines 238-239 – Yes, they could but from section 2.1. it is unclear which was vegetative stage of plants. Also, I assume that post-harvest conditions were the same for all samples. Thus this sentence might be irrelevant.
Line 240 – please delete furthermore, it is redundant there
Line 242 – than reported
Lines 244-245 – Flavonoids are related to plant for the region not region itself, rephrase the sentence.
Lines 262-264 – The same comment as above.
Reference 7 is not correct, it is a copy of reference 4, please correct this.
Figure 3. Candida Albicans. If possible please increase the quality of figures.
Figure 4. Please use dots as decimal separators.
Line 340 – reference is incorrect form
Lines 376-377 – same comment as for lines 220-224, annual rainfall for the harvest year should be checked before giving this statement.
Section 3.7 - Many sources indicate that PCA, as other multivariate statistical methods, should not include dependent variables. Values of TF, Tanin, TP and antioxidant activity depend on the content of the individual polyphenols, so higher concentrations of individual polyphenols obviously lead to higher TF, Tanin and TP. Thus all dependent variables should be removed from PCA.
https://www.graphpad.com/guides/prism/latest/statistics/stat_qa_pca.htm
https://www.researchgate.net/post/Should_I_insert_dependent_variable_into_Principal_Component_Analysis
Line 388 – Supplementary material is missing
Author Response
Dear Editor
You find as an attachment the response to reviewer
Best Regards

Round 2
Reviewer 1 Report
The authors have addressed all the issues. The manuscript is suitable for publication.
Reviewer 2 Report
Authors significantly improved the manuscript and it can be accepted for publication.